# Sustainability in Education: A Scale on Perceptions of Organisational Discipline Related to the COVID-19 Period

**Rasiha Yerel** [1,*], **Gokmen Dagli** [1,2,*], **Fahriye Altinay** [1,*], **Ebba Ossiannilsson** [3], **Mehmet Altinay** [2] and **Zehra Altinay** [1,*]

1   Institute of Graduate Studies, Faculty of Education, Social Research and Development, Near East University, Nicosia 99138, Cyprus
2   Faculty of Business, University of Kyrenia, Kyrenia 99320, Cyprus; mehmet.altinay@kyrenia.edu.tr
3   Swedish Association for Distance Education (SADE), International Council for Open and Distance Education (ICDE), OER Advocacy Committee, 222 35 Lund, Sweden; ebba.ossiannilsson@gmail.com
*   Correspondence: yerelrasiha@gmail.com (R.Y.); gokmen.dagli@neu.edu.tr (G.D.); fahriye.altinay@neu.edu.tr (F.A.); zehra.altinaygazi@neu.edu.tr (Z.A.); Tel.: +90-533-834-7438 (R.Y.)

**Abstract:** Higher education institutions are the most important institutions that form the basis of societies. The devoted students, academic and administrative employees enlighten the future and keep these institutions alive. Employees of higher education institutions have to teach a discipline with the order existing in their organisation and their relations with each other. The purpose of this study is to determine using the "organisational discipline" perceptions of higher education employees in different situations and determine the differences according to their socio-demographic characteristics. In the literature review of this study, no scale related to the perception of organisational discipline was found. For this purpose, a scale consisting of six sub-dimensions that will cover the perception of organisational discipline was prepared. The validity and reliability of the prepared scale were determined by conducting a study. COVID-19 pandemic affected the world in all sectors throughout, a prepared scale was applied to 357 higher education institution academics who has been teaching online for a long time. Forward-looking sustainable higher education strategy, the discipline scale plays an essential role in the organisation. In this period, the perceptions of organisational discipline were tried to be determined by considering their current situation. The data obtained in the research were applied independent sample *t*-test and analysis of variance (ANOVA); *T*-test, one of the post-hoc tests, was used as further analysis. The study determined that if there is significant difference in perceptions of organisational discipline related to gender, age, nationality, educational status, professional seniority, academic title and working style.

**Keywords:** education 4.0; higher education; organisational discipline; sustainability in education; COVID-19

## 1. Introduction

### 1.1. New Actions for Transforming the Future of Higher Education after COVID-19

Higher education institutions were globally affected by the virus that emerged in China in late 2020, shortly becoming a pandemic. Worldwide, most countries turned into emergency remote education (ERE) [1], although they began to turn into an academic framework for a more sustainable online distance education. Turkish Republic of Northern Cyprus (TRNC) also switched to distance education without wasting time. Turkey also began on 18 March 2020, with a press release announcing those distance education activities. According to these fields, universities with ready-made infrastructure were able to start online distance education. In contrast, others had to prepare the necessary configurations to start distance education and switch to distance education quickly with appropriate infrastructure support. Studies were also to be carried out to train new people for people who would support distance education in human resources. The course contents would

be rewritten to make them ready for the use of all universities. All this work together with the Council of Higher Education to support universities, Turkey Radio and Television Corporation, will publish the lessons [2]. Washington University, then Harvard University, followed by universities in California and New York, decided to close the schools in the United States on 6 March 2020. More than 1100 colleges and universities in 50 states have cancelled face-to-face classes or switched to online education only [3]. In Southeast Asia, due to COVID-19, many schools and universities have changed to online learning to prevent further transmission of the disease [4]. According to [4], there is no policy determined in Southeast Asian countries for this process. Italy was one of the most affected countries by COVID-19 after China and had to stop education rapidly. The training was suspended on March 4. It was said that this break would take 10 days, but the expected breakthrough was not [5]. They were ensuring the sustainable continuation of education expected in all countries and turning to online applications.

With the global pandemic, with the awareness that education can be done anywhere, higher education institution employees began to adapt to distance education online. When the operation changed and the houses became "home-offices," people understood the difference between working in the old days and the home environment better. They provided the opportunity to better review and empathise with some complicated rules for people in business life. In this process, higher education institutions continued to supervise the employees of the institutions in terms of management and discipline and to follow them to ensure development. The word discipline it means to teach, to educate [6]. Behaviors and efforts to increase the awareness and willingness of the employees of the organization to comply with all the rules are effective in maintaining the disciplinary management of the organization [7]. Personal situations, which are among the disciplinary factors that affect performance in the organization, Skill level, competence, motivation, commitment affect the work situation and can be achieved with discipline [8]. It is seen that higher education institutions are also affected in terms of organizational discipline in crisis situations that occur in organizations. While the resulting confusion creates a crisis situation, this is a situation in normal reality as it is life itself [9]. Although there are some changes in the individuals affected by such situations, the discipline understanding of the managers and employees in the organization is an important issue for the sustainability of organizational discipline.

The research was carried out by developing a scale and applying it to the employees as academicians of higher education institutions to determine the organisation's discipline perception regarding the sustainability of education. The aim was to contribute to the sustainability of education with the scale developed to determine the organisational discipline perception of the employees in the higher education institution during the distance education period. In the COVID-19 period, the work done by institutions in terms of distance education is essential, and it is stated that the effects of distance education will continue with these studies, even if face-to-face education is started. Online studies at universities will be accelerated; technologically, schools will compensate for their shortcomings and go to a regular organisation; all institutions will continue to use the system they have established during the pandemic period in times of crisis [10]. A well-structured distance education model in the crisis period for sustainable education will create many opportunities for students and teachers. In this period, the successful effect of the person who will manage the institution and increase teachers' intrinsic motivation will make a sustainable contribution to education [11].

### 1.2. The Importance of Organisational Discipline in Higher Education in the New Management Model

In the new management model that started to emerge with the pandemic period, the importance of organisational discipline in higher education has emerged as models associated with an effective and efficient internal control system [12]. An attempt was made to conduct online audits in these new internal control systems.

School closures have also forced countries to innovate to maintain their education systems. The education system was started to be questioned, gaps were identified and new ways were sought by determining that individuals could be affected by these gaps. A new road has begun to take shape globally. In this period, countries have tried to maintain the education and management dimension in a quality way by focusing on digital online learning tools to continue education and quality education [13].

Discipline is the power that enables human resources to believe and willingly comply with the institution's rules and act by the institutional order [14]. For the individual, discipline is the ability of the individual to control himself without being influenced by others. In terms of organisations, discipline can be defined in two ways. The first of these is criminal discipline. The other is non-criminal discipline. According to criminal discipline, discipline is the use of penalties to prevent unwanted behaviour. According to the non-criminal discipline, discipline is to constitute the presence of personnel who willingly abide by the rules and regulations and as a result to create an organisational climate and attitude [15]. Various dimensions such as climate, attitude, and commitment that emerge from discipline in organisations carry the organisation forward. During the COVID-19 pandemic period, sustainable education employees in higher education institutions are provided with the knowledge of the rules, punishments, rewards, or procedures regarding organisational discipline. With the integration of new technologies into the system, a change has begun in higher education. Many studies that can be planned to be carried out in five years showed themselves in universities in a short time with a sense of commitment and responsibility. Although an epidemic affecting the world is expressed as a disaster, it may be the beginning of some things [16].

With the increasing importance of higher education, politicians and state bodies from universities can be pretty high. University administrators and academic staff should not avoid fulfilling their responsibilities due to this situation and should act with this awareness. Suppose there are any problems regarding the functioning of the current system and sanctions. In that case, these should be revealed and efforts should be continued to become a "University Island" by eliminating them as soon as possible. With the Higher Education Law enacted on 13 December 2005 in the Turkish Republic of Northern Cyprus, where there are twenty universities, a series of general rules were introduced for the administrators appointed to universities to establish their internal order [17]. However, the organisation's discipline is considered necessary in terms of whether the expectations in the field of implementation are met and continuity within the organisation can be achieved. In this respect, the sub-dimensions of the organisational discipline can be determined and handled. The perceptions of higher education institution employees on this issue can be taken into account. While universities are meant as "organisation" in the concept of "organisational discipline,"; With the "organisational discipline," the set of rules that the employees should follow in the administration dimension of universities is expressed. It is essential to establish discipline in the organisation. However, some problems may arise in some chaotic situations. COVID-19 pandemic was one of these situations. Preparing schools for the crisis period under the theme of alternative education structures in the reports prepared with the pandemic regarding education, developing distance education programs, and creating new online learning platforms are among the opportunities in this crisis [18].

In the literature review of this study, no scale related to the perception of organisational discipline was found. For this purpose, a scale consisting of six sub-dimensions that will cover the perception of organisational discipline was prepared. The sub-dimensions of the scale are "Internal communication and organisational rules," "Commitment to the Organisation," "Organisational incentive structure," "Organisation and the place of punishment," "Organisation regulation and management authority," "Organisational climate and control of individuals." For the questions representing the sub-dimensions, a question bank was created in line with the information obtained from the literature review, and the scale was designed by taking expert opinions. It is aimed to fill the gap in the field

with a valid and reliable organisational discipline perception scale prepared according to these sub-dimensions. With the prepared valid and reliable scale, organizational discipline perceptions of higher education employees are determined and their differentiation levels are revealed according to their socio-demographic characteristics. To achieve this aim, the following research questions were created and answers were sought.

Research Question (RQ) 1: Is the scale developed to determine the organisational discipline perception of higher education employees in sustainable education in the COVID-19 period sufficient in terms of (a) validity and (b) reliability?

Research Question (RQ) 2: Is there any significant difference between the level of differentiation in the scores of academicians obtained from the organisational discipline scale according to their socio-demographic characteristics?

Research Question (RQ) 3: What is the distribution of the academicians participating in the organisational discipline scale according to their socio-demographic variables;

(a) gender, (b) age, (c) nationality (d) educational status (e) professional seniority, (f) academic title and (g) study type.

## 2. Materials and Methods

The study was based on a survey for the quantitative data, which is one of the techniques generally used in the scanning model, was used. The scale was prepared in a 5-point Likert type. Accordingly, the participants answered the questionnaire as strongly disagree, disagree, neither agree nor disagree, agree, strongly agree. The questionnaires were delivered to a group of 357 randomly selected personnel working in higher education institutions during the pandemic period.

In the study, the organisational discipline scale was used with personal information form as a quantitative data collection tool in higher education institutions. A question bank was created by making a literature review on the subject and determining which questions or items would be appropriate for the scale subject. When designing questions to be understandable, care was taken. Expert opinion was requested for the questions in the created question bank. Experts on the necessity, clarity, and specificity of questions made their assessment. The scale, which was shaped according to expert opinion, was applied to the sample group. For the validity and reliability study of the draft Organisational Discipline Scale developed by the researcher based on the literature review, the opinions of Turkish teachers and experts of the subject and the construct validity of the scale were examined first. While examining the construct validity of the Organisational Discipline Scale, Explanatory Factor Analysis (EFA) and Confirmatory Factor Analysis (CFA) was used to determine the scale's factor structure.

*Data Analysis*

Statistical Package for Social Sciences (SPSS) and IBM AMOS 21.0 software were used to analyse the research data quantitatively.

In the validity-reliability study of the Organisational Discipline Scale, the construct validity and reliability values were examined. The scale was prepared by determining appropriate questions by applying exploratory factor analysis and confirmatory factor analysis within the scope of construct validity. Cronbach Alpha test, Split-Half test and item-total correlations were used to examine the reliability of the scale.

To determine the socio-demographic characteristics of the academicians participating in the study, frequency analysis was applied and descriptive statistics regarding the scores they got from the Organisational Discipline Scale were shown.

The normal distribution of the scores of the academicians from the Organisational Discipline Scale was decided by looking at the Kolmogorov-Smirnov test, the QQ plot graph, the skewness and kurtosis values, and it was determined that they fit the normal distribution. For this reason, parametric hypothesis tests were used to compare the participants' scores from the Organisational Discipline Scale according to their socio-demographic characteristics. An independent sample *t*-test was used when comparing the scores of the

Organisational Discipline Scale according to the gender, education level and working style of the academicians. Analysis of variance (ANOVA) was used to compare the Organisational Discipline Scale scores according to the age group, nationality, professional seniority and title of the academicians and *t*-test. One of the post-hoc tests was used for further analysis.

## 3. Results

### 3.1. Validity-Reliability Study of the Organisational Discipline Scale

For the validity-reliability study of the draft form of the Organisational Discipline Scale, which the researcher developed by taking the literature review and the experts' opinions, the scale's construct validity was examined first. Explanatory Factor Analysis (EFA) and Confirmatory Factor Analysis (CFA) were used to examine the construct validity of the Organisational Discipline Scale, and the findings are given below.

#### 3.1.1. Structure Validity Exploratory Factor Analysis (EFA)

For the exploratory factor analysis to be applied, the scale items should conform to the multivariate normal distribution. For this reason, the compliance of the draft form of the Organisational Discipline Scale to the multivariate normal distribution was examined and it was determined that it was compatible with the normal distribution. After the normality assumption required for the Explanatory Factor analysis was provided, the Kaiser-Meer-Olkin (KMO) coefficient and Bartlett's test were applied to determine the suitability of the data for EFA. It was observed that the draft form of the Organisational Discipline Scale had a Kaiser-Meer-Olkin coefficient of 0.952. The KMO coefficient gives information about whether the data matrix is suitable for factor analysis and the suitability of the data structure for factor extraction. For factorization, KMO is expected to be higher than 0.60. When Bartlett's Test of Sphericity results was examined, it was determined that the calculated chi-square value of the test was 3824,592 and this value was statistically significant ($p < 0.05$). Bartlett's test examines whether there is a relationship between variables were based on partial correlations [19].

The assumptions required for applying factor analysis to the data set, such as normality, high Kaiser–Meer–Olkin value and Bartlett's Test of Sphericity value, have been provided, showing that exploratory factor analysis can be applied draft form of the Organisational Discipline Scale. Before evaluating the exploratory factor analysis results, the Scree Plot chart is given in Figure 1 to get an idea about the factor structure of the draft form of the Organisational Discipline Scale.

The exploratory factor analysis scree plot chart of the draft form of the Organisational Discipline Scale is shown in Figure 1. It was seen that the breaking point of the graph was at the sixth factor and from this factor the graph became flat. This indicates that the scale has a six-factor structure.

The Principal Components method was used in the exploratory factor analysis to determine the factor structure of the Organisational Discipline Scale, and the varimax transformation was applied to the data set.

The core values obtained as a result of the exploratory factor analysis applied to the Organisational Discipline Scale and the variances they explain are shown in Table 1.

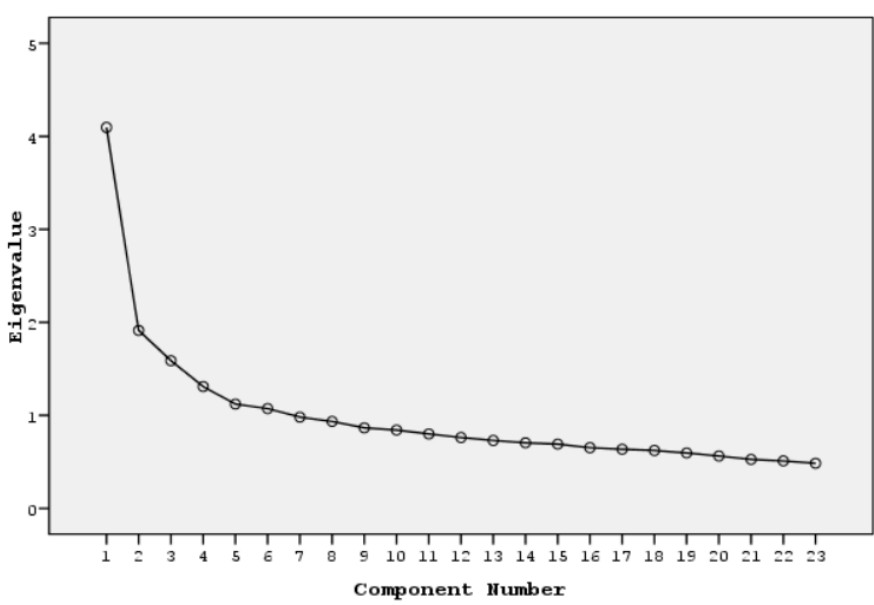

**Figure 1.** Exploratory factor analysis of the draft form of the Organisational Discipline Scale Scree Plot Graph.

**Table 1.** Results of the Exploratory Factor Analysis of the Organisational Discipline Scale.

| Factor | Core Values | | | Rotated Sum of Squares | | |
|---|---|---|---|---|---|---|
| | Core Value | Disclosed Variance | Cumulative Variance (%) | Core Value | Disclosed Variance | Cumulative Variance (%) |
| Factor 1 | 3.97 | 13.24 | 13.24 | 3.45 | 11.50 | 11.50 |
| Factor 2 | 2.63 | 8.76 | 22.00 | 2.65 | 8.84 | 20.34 |
| Factor 3 | 2.17 | 7.23 | 29.23 | 2.43 | 8.09 | 28.43 |
| Factor 4 | 2.02 | 6.73 | 35.97 | 2.08 | 6.95 | 35.38 |
| Factor 5 | 1.91 | 6.38 | 42.35 | 2.04 | 6.79 | 42.17 |
| Factor 6 | 1.76 | 5.88 | 48.23 | 1.82 | 6.06 | 48.23 |

Table 1. When examined, according to the results of exploratory factor analysis; It was determined that there are six factors with a core value (λ) above 1 in the Organisational Discipline Scale. The biggest of these factors is the core value (λ = 3.97), and this factor can explain 13.24% of the change (variance) in the Organisational Discipline Scale. The core value of the second factor in the scale is (λ = 2.63) and this factor alone can explain 8.76% of the change in the Organisational Discipline Scale. The core value of the third factor is (λ = 2.17) and the third factor alone can explain 7.23% of the change in the Organisational Discipline Scale. The core value of the fourth factor in the Organisational Discipline Scale (λ = 2.02) was found. The fourth factor alone explains 6.73% of the change in the scale. The core values of the fifth and sixth factors in the scale were found to be λ = 1.91 and λ = 1.76, respectively. The fifth factor alone explains 6.38% of the variance in the Organisational Discipline Scale, while the sixth factor alone can explain 5.88% of the variance. The six-factor structure of the Organisational Discipline Scale can explain 48.23% of the total change.

In Table 1 Results of the Exploratory Factor Analysis of the Organisational Discipline Scale items are shown.

Table 2 The factor structure of the scale obtained from the exploratory factor analysis applied to the Organisational Discipline Scale and the factor loadings of the scale items are shown.

**Table 2.** Factor structure and factor loadings of the Organisational Discipline Scale.

|  | Factor 1 | Factor 2 | Factor 3 | Factor 4 | Factor 5 | Factor 6 |
|---|---|---|---|---|---|---|
| Item 37 | 0.688 |  |  |  |  |  |
| Item 38 | 0.650 |  |  |  |  |  |
| Item 20 | 0.624 |  |  |  |  |  |
| Item 34 | 0.620 |  |  |  |  |  |
| Item 44 | 0.619 |  |  |  |  |  |
| Item 42 | 0.612 |  |  |  |  |  |
| Item 18 | 0.597 |  |  |  |  |  |
| Item 9 | 0.427 |  |  |  |  |  |
| Item 31 | 0.423 |  |  |  |  |  |
| Item 43 | 0.379 |  |  |  |  |  |
| Item 46 |  | 0.907 |  |  |  |  |
| Item 45 |  | 0.902 |  |  |  |  |
| Item 47 |  | 0.900 |  |  |  |  |
| Item 8 |  |  | 0.845 |  |  |  |
| Item 12 |  |  | 0.821 |  |  |  |
| Item 11 |  |  | 0.663 |  |  |  |
| Item 25 |  |  | 0.561 |  |  |  |
| Item 40 |  |  |  | 0.899 |  |  |
| Item 39 |  |  |  | 0.881 |  |  |
| Item 48 |  |  |  | 0.340 |  |  |
| Item 10 |  |  |  | 0.336 |  |  |
| Item 49 |  |  |  |  | 0.787 |  |
| Item 21 |  |  |  |  | 0.772 |  |
| Item 30 |  |  |  |  | 0.420 |  |
| Item 22 |  |  |  |  | 0.383 |  |
| Item 27 |  |  |  |  | 0.354 |  |
| Item 7 |  |  |  |  | 0.349 |  |
| Item 6 |  |  |  |  |  | 0.879 |
| Item 5 |  |  |  |  |  | 0.871 |
| Item 1 |  |  |  |  |  | 0.325 |

Table 2. When examined; In the exploratory factor analysis performed from the Organisational Discipline Scale, the draft form of which consists of 53 items, it was determined that the scale was reduced to 30 items as a result of the exclusion of a total of 23 items with a factor load of less than 0.30. Item 37, Item 38, Item 20, Item 34, Item 44, Item 42, Item 18, Item 9, Item 31 and Item 43 in the scale constitute the first factor and this factor is named "Internal communication and organisational rules".

The second factor in the scale consists of Item 46, Item 45 and Item 47 and is named "Commitment to the organisation".

The third factor consists of items 8, 12, 11 and 25 and named "Intra-organisational incentive structure".

Item 40, Item 39, Item 48 and Item 10 in the Organisational Discipline Scale are loaded in the fourth factor and the name of this factor is "belonging to the organisation and the place of punishment."

The fifth factor is named "Organisation regulation and managerial authority," and Items 49, Item 21, Item 30, Item 22, Item 27 and Item 7 in the scale have been loaded under this factor.

The sixth factor, which is the last factor in the Organisational Discipline Scale, consists of Item 6, Item 5 and Item 1 and has been named "Organisational climate and control of individuals".

Items indicated by numbers in the sub-dimensions are included in Appendix A.

3.1.2. Structure Validity Confirmatory Factor Analysis (CFA)

After the factor structure of the Organisational Discipline scale was revealed, confirmatory factor analysis was applied to confirm the suitability of the factor structure of the

scale and to examine the determination of the relationships between factors. Confirmatory factor analysis is an extension of the explanatory factor analysis. While explanatory factor analysis explains the factor structure of a measurement tool, confirmatory factor analysis is used to test whether the relationship between EFA and the factors determined is sufficient, the variables are related to each other, whether the factors are independent of each other and whether the determining factors are sufficient to explain the established model [20].

When the confirmatory factor analysis index values of the Organisational Discipline Scale given in Table 3 are examined, it is determined that $\chi^2$/sd is 1.61. This situation indicates that the Organisational Discipline Scale has a perfect fit in terms of $\chi^2$/sd. According to Kline in 2005 [21]. A value of $\chi^2$/sd below 3 indicates that it is a perfect fit, and between 3 and 5 indicates an acceptable fit.

**Table 3.** Organisational discipline scale confirmatory factor analysis goodness of fit index values.

| Index | Value | Boundary Value | Fit |
| --- | --- | --- | --- |
| $\chi^2$/sd | 1.61 | 3–5 | Excellent |
| Goodness Fit Index (GFI) | 0.90 | 0.90–0.95 | Acceptable |
| Normed Fit Index (NFI) | 0.84 | 0.90–0.95 | Bad |
| Comparative Fit Index (CFI) | 0.93 | 0.90–0.95 | Acceptable |
| The Root Mean Square Error of Approximation(RMSEA) | 0.04 | 0.5–0.8 | Excellent |

A goodness of fit index (GFI) index between 0.95 and 1.00 indicates the presence of a perfect fit, while between 0.90 and 0.95 indicate an acceptable fit [22]. The Goodness Index (GFI) index determined for the Organisational Discipline Scale was found to be 0.90. The scale was found to have an acceptable fit in terms of the Well-Being Adaptation Index.

According to the confirmatory factor analysis results of the Organisational Discipline Scale, the Normalized Fit Index (NFI) value of the scale was found to be 0.84. According to [23], the limit value determined for the Normed Fit Index is between 0.90 and 1.00. The NFI value is expected to be between the specified limit values, and this indicates acceptable compliance. In this respect, it was determined that the NFI value determined for the Organisational Discipline Scale was below the specified limit value and the measurement tool was not compatible with NFI.

Among the critical values determined for the Comparative Fit Index (CFI), the range 0.95–1.00 indicates the presence of a good fit, and the range 0.90–0.95 indicates the presence of an acceptable fit [23]. The Comparative Fit Index value determined for the Organisational Discipline Scale is 0.93. This value shows that the Organisational Discipline Scale has an acceptable fit for the Comparative Fit Index.

The Approximate Root Mean Square Error (RMSEA) value of the Organisational Discipline Scale, which is another index of goodness of fit obtained as a result of the confirmatory factor analysis, was found to be 0.04. According to [24], RMSEA value between 0.00 and 0.05 indicates perfect fit, and between 0.05 and 0.08 indicates acceptable fit. Accordingly, it was determined that the Organisational Discipline Scale has a perfect fit in terms of RMSEA.

According to the confirmatory factor analysis of the Organisational Discipline Scale: The 37th item, 38th item, 20th item, 34th item, 44th item, 42nd item, 18th item, 9th item and 43rd item of organisational communication and organisational rules Commitment to the organisation sub-dimension of the organisational incentive structure item 8, 12, 11 and 25, belonging to the organisation sub-dimension, and the punitive position of item 40, 39, 48 and 10, Organisation regulating and managing authority sub-dimension Article 49, item 21, item 30 consists of item 22, 27 and 7, and the organisational climate and control of individuals sub-dimension consist of item 6, 5 and 1.

In line with the findings shown above, it was determined that the confirmatory factor analysis model of the Organisational Discipline Scale was suitable for the goodness of fit index and that the 30-item form of the Organisational Discipline Scale was preserved without removing any items from the final form determined by the exploratory factor

structure. According to the findings obtained from EFA and CFA, it was determined that the Organisational Discipline Scale has a six-factor structure with 30 items and is a valid measurement tool.

### 3.1.3. Reliability

To examine the reliability of the Organisational Discipline Scale, Cronbach alpha test and Split Half test were performed, respectively, and item-total correlations in the scale were examined.

The alpha coefficients of the Cronbach's Alpha test results of the Organisational Discipline Scale are shown in Table 4.

**Table 4.** Cronbach alpha test results of the organisational discipline scale.

|  | Alfa |
|---|---|
| Internal communication and organisational rules | 0.776 |
| Commitment to knitting | 0.911 |
| Intra-organisational incentive structure | 0.743 |
| Belonging to the organisation and place of punishment | 0.634 |
| Organisation regulation and managing authority | 0.721 |
| Organisational climate and control of individuals | 0.711 |
| Organisational Discipline Scale | 0.718 |

As seen in Table 4 above, the alpha reliability coefficient of the overall Organisational Discipline Scale was found as 0.718. According to [25], if the Cronbach Alpha reliability coefficient is above 0.70, the measurement tool is reliable. When the Cronbach Alpha values of the sub-dimensions in the scale are examined; It was found to be 0.776 for intra-organisational communication and organisational rules, 0.911 for commitment to the Organisation, 0.743 for the structure of incentives within the organisation, 0.634 for belonging to the Organisation and the place of punishment, 0.721 for the organisational regulation and executive authority, and 0.718 for the organisational climate and control of individuals.

The results of the Split-Half test of the Organisational Discipline Scale are shown in Table 5.

**Table 5.** Results of the split-half test of the organisational discipline scale.

| Split-Half Test | Coefficients |
|---|---|
| First Half Cronbach Alfa (15 items) | 0.702 |
| Second Half Cronbach Alfa (15 items) | 0.757 |
| Correlation between halves | 0.559 |
| Spearman-Brown Coefficient | 0.729 |
| Guttman Split-Half Coefficient | 0.716 |

Table 5. When examined, the Cronbach's alpha value of the first part with 15 items of the Organisational Discipline Scale was 0.702 and the Cronbach alpha value of the second half of 15 items was 0.757. The correlation between the two halves of the Organisational Discipline Scale with 15 items was r = 0.559, the Spearman-Brown coefficient was 0.729 and the Guttman Split-Half coefficient was 0.716.

When the item-total correlations of the data were examined in Table 6, it was found that the item-total correlations in the scale varied between 0.293 and 0.881. The item with the highest item-total correlation was 40 and the lowest item 9.

**Table 6.** Organisational discipline scale item-total correlations.

| | Correlation |
|---|---|
| Item 1 | 0.322 |
| Item 5 | 0.576 |
| Item 6 | 0.565 |
| Item 7 | 0.344 |
| Item 8 | 0.638 |
| Item 9 | 0.283 |
| Item 10 | 0.375 |
| Item 11 | 0.311 |
| Item 12 | 0.613 |
| Item 18 | 0.293 |
| Item 20 | 0.317 |
| Item 21 | 0.622 |
| Item 22 | 0.346 |
| Item 25 | 0.316 |
| Item 27 | 0.388 |
| Item 30 | 0.391 |
| Item 31 | 0.334 |
| Item 34 | 0.333 |
| Item 37 | 0.414 |
| Item 38 | 0.350 |
| Item 39 | 0.879 |
| Item 40 | 0.881 |
| Item 42 | 0.392 |
| Item 43 | 0.321 |
| Item 44 | 0.452 |
| Item 45 | 0.734 |
| Item 46 | 0.743 |
| Item 47 | 0.664 |
| Item 48 | 0.383 |
| Item 49 | 0.619 |

According to the results mentioned above, it has been determined that the Organisational Discipline Scale is a valid and reliable measurement tool.

*3.2. Findings Regarding the Descriptive Study of the Organisational Discipline Scale*

In this section, the socio-demographic characteristics of the academicians are included in the research. Findings regarding the scores obtained from the Organisational Discipline Scale and comparing the scale scores according to socio-demographic characteristics are included.

The findings regarding the distribution of the academicians participating in the research according to their socio-demographic characteristics are shown in Table 7.

**Table 7.** Socio-demographic characteristics of academics (*n* = 357).

| | Number (*n*) | Percent (%) |
|---|---|---|
| **Sex** | | |
| Female | 181 | 50.70 |
| Male | 176 | 49.30 |
| **Age** | | |
| Under 30 | 76 | 21.29 |
| 30–39 years old | 120 | 33.61 |
| 40–49 years old | 68 | 19.05 |
| 50 years and older | 93 | 26.05 |

**Table 7.** *Cont.*

|  | Number (*n*) | Percent (%) |
|---|---|---|
| **Nationality** |  |  |
| TRNC | 133 | 37.25 |
| TR | 110 | 30.81 |
| TRNC + TR | 80 | 22.41 |
| Other | 34 | 9.52 |
| **Education Status** |  |  |
| Post Graduate | 85 | 23.81 |
| Doctorate | 272 | 76.19 |
| **Professional Seniority** |  |  |
| 5 years and below | 101 | 28.29 |
| 6–15 years | 102 | 28.57 |
| 16 years and above | 154 | 43.14 |
| **Title** |  |  |
| Expert | 86 | 24.09 |
| Doctor | 30 | 8.40 |
| Assistant Assoc. | 135 | 37.82 |
| Associate professor | 45 | 12.61 |
| Professor | 61 | 17.09 |
| **Work Type** |  |  |
| Full time | 262 | 73.39 |
| Part-time | 95 | 26.61 |

Table 8. The scores of the academicians from the organisational discipline scale are shown below.

**Table 8.** The scores of the academicians from the organisational discipline scale (*n* = 357).

|  | *n* | $\bar{x}$ | s | Bottom | Top |
|---|---|---|---|---|---|
| Internal communication and organisational rules | 357 | 28.78 | 5.96 | 14 | 40 |
| Commitment to knitting | 357 | 8.29 | 2.84 | 3 | 15 |
| Intra-organisational incentive structure | 357 | 14.22 | 3.05 | 5 | 20 |
| Belonging to the organisation and place of punishment | 357 | 14.55 | 2.81 | 5 | 20 |
| Organisation regulation and managing authority | 357 | 20.20 | 3.40 | 9 | 29 |
| Organisational climate and control of individuals | 357 | 11.05 | 1.97 | 4 | 15 |

Table 9 shows the findings obtained from the independent sample *t*-test applied to compare the academicians' scores in the study from the Organisational Discipline Scale according to their gender.

**Table 9.** Comparison of academicians' scores from the organisational discipline scale according to their gender (*n* = 357).

|  | Sex | *n* | $\bar{x}$ | s | t | p |
|---|---|---|---|---|---|---|
| Internal communication and organisational rules | Female | 181 | 28.52 | 5.95 | −0.833 | 0.405 |
|  | Male | 176 | 29.05 | 5.99 |  |  |
| Commitment to knitting | Female | 181 | 8.23 | 2.98 | −0.381 | 0.704 |
|  | Male | 176 | 8.35 | 2.69 |  |  |
| In-house | Female | 181 | 14.25 | 2.93 | 0.241 | 0.809 |
|  | Male | 176 | 14.18 | 3.18 |  |  |
| Incentive structure | Female | 181 | 14.58 | 2.76 | 0.173 | 0.863 |
|  | Male | 176 | 14.53 | 2.88 |  |  |
| Belonging to the organisation and the place of punishment | Female | 181 | 21.52 | 3.33 | 8.080 | 0.000 |
|  | Male | 176 | 18.85 | 2.90 |  |  |
|  | Female | 181 | 11.64 | 1.93 | 5.984 | 0.000 |
|  | Male | 176 | 10.44 | 1.83 |  |  |

According to the gender of the academicians included in the study, it was determined that there was no statistically significant difference between the scores of the organisational communication and organisational rules, commitment to the organisation, organisational incentive structure, belonging to the organisation and the place of punishment, and the organisational regulation and administrative authority sub-dimension ($p > 0.05$). The scores obtained by male and female academicians in these sub-dimensions are similar.

It was determined that the difference between the scores of the academicians included in the study from the organisational climate and control of individuals sub-dimension according to their gender was statistically significant ($p < 0.05$). The scores of female academicians in the organisational climate and control of individuals sub-dimension were higher than male academicians.

Table 10 shows the results of the analysis of variance applied to compare the scores of the academicians from the Organisational Discipline Scale according to the age group.

**Table 10.** Comparison of the scores of the academicians from the organisational discipline scale according to the age group ($n = 357$).

| | Age | $n$ | $\bar{x}$ | s | Bottom | Top | F | $p$ | Variation |
|---|---|---|---|---|---|---|---|---|---|
| To knit belonging and place of punishment | Under 30 | 76 | 21.89 | 3.10 | 15 | 29 | 195.858 | 0.000 * | 1–3 |
| | 30–39 years old | 120 | 26.68 | 4.59 | 14 | 38 | | | 1–4 |
| | 40–49 years old | 68 | 32.91 | 3.22 | 24 | 40 | | | 2–3 |
| | 50 years and older | 93 | 34.09 | 3.01 | 23 | 39 | | | 2–4 |
| Organisation regulation | Under 30 | 76 | 7.51 | 2.40 | 3 | 15 | 7.327 | 0.000 * | 1–4 |
| | 30–39 years old | 120 | 8.08 | 2.78 | 3 | 15 | | | |
| | 40–49 years old | 68 | 8.01 | 2.75 | 3 | 15 | | | |
| | 50 years and older | 93 | 9.39 | 3.03 | 3 | 15 | | | |
| and manager authority | Under 30 | 76 | 12.22 | 3.41 | 5 | 20 | 17.331 | 0.000 * | 1–2 |
| | 30–39 years old | 120 | 14.90 | 2.45 | 8 | 19 | | | 1–3 |
| | 40–49 years old | 68 | 15.21 | 2.47 | 8 | 19 | | | 1–4 |
| | 50 years and older | 93 | 14.24 | 3.10 | 6 | 20 | | | |
| Organisational climate and persons | Under 30 | 76 | 14.22 | 2.65 | 8 | 20 | 0.467 | 0.705 | |
| | 30–39 years old | 120 | 14.63 | 2.59 | 8 | 19 | | | |
| | 40–49 years old | 68 | 14.72 | 3.12 | 6 | 20 | | | |
| | 50 years and older | 93 | 14.60 | 3.00 | 5 | 20 | | | |
| To knit belonging and place of punishment | Under 30 | 76 | 19.66 | 3.57 | 9 | 26 | 0.845 | 0.470 | |
| | 30–39 years old | 120 | 20.32 | 3.26 | 13 | 29 | | | |
| | 40–49 years old | 68 | 20.31 | 3.55 | 12 | 26 | | | |
| | 50 years and older | 93 | 20.42 | 3.30 | 11 | 27 | | | |
| | Under 30 | 76 | 11.03 | 2.08 | 4 | 15 | 0.143 | 0.934 | |
| | 30–39 years old | 120 | 11.12 | 2.05 | 6 | 15 | | | |
| | 40–49 years old | 68 | 11.09 | 1.94 | 7 | 15 | | | |
| | 50 years and older | 93 | 10.95 | 1.83 | 6 | 15 | | | |

* $p < 0.05$.

According to the age group, a statistically significant difference was found between the scores of the academicians in the organisational communication and organisational rules sub-dimension ($p < 0.05$). The scores of academicians under the age of 30 and in the age group 30–39 were lower than those in the 40–49 age group and the age group 50 and over.

It was determined that the difference between the scores of the academicians included in the study from the sub-dimension of commitment to organisation according to the age group was statistically significant ($p < 0.05$). The scores obtained by the academicians under the age of 30 from the sub-dimension of commitment to the organisation are lower than those of the academicians 50 years and over.

It was determined that the scores of the academicians from the sub-dimension of the organisational incentive structure differed statistically significantly according to age groups

($p < 0.05$). The scores of the academicians under the age of 30 from the intra-organisational incentive structure sub-dimension were lower than the academicians in other age groups.

It was observed that there was no statistically significant difference between the scores of the academicians in the sub-dimensions of belonging to the organisation and the place of punishment, organisational regulation and executive authority, organisational climate, and control of individuals ($p > 0.05$).

The results of variance analysis regarding the comparison of the scores of the academicians included in the study from the Organisational Discipline Scale according to their nationalities are given in Table 11. There is a statistically significant difference between the scores of the academicians participating in the study from the sub-dimension of intra-organisational communication and organisational rules ($p < 0.05$). Only the academicians with TRNC nationality are lower than the academicians with dual nationality, TRNC and TR, and academicians from other countries in the sub-dimension of intra-organisational communication and organisation rules. Besides, the scores obtained by only Turkish nationals from the sub-dimension of intra-organisational communication and organisational rules were found to be lower than the academicians of other countries.

**Table 11.** Comparison of academicians' scores from the organisational discipline scale according to their nationality ($n = 357$).

| | Nationality | $n$ | $\overline{x}$ | s | Bottom | Top | F | $p$ | Variation |
|---|---|---|---|---|---|---|---|---|---|
| In-organisation Contact and organisation rules | TRNC | 133 | 27.38 | 5.26 | 15 | 39 | 8.371 | 0.000 * | 1–3 |
| | TR | 110 | 28.33 | 6.58 | 14 | 40 | | | 1–4 |
| | TRNC + TR | 80 | 30.35 | 5.30 | 18 | 39 | | | 2–4 |
| | Other | 34 | 32.03 | 6.08 | 17 | 39 | | | |
| To knit | TRNC | 133 | 8.45 | 3.01 | 3 | 15 | 0.517 | 0.671 | |
| | TR | 110 | 8.03 | 2.72 | 3 | 15 | | | |
| | TRNC + TR | 80 | 8.29 | 2.72 | 3 | 15 | | | |
| | Other | 34 | 8.50 | 2.86 | 3 | 15 | | | |
| loyalty | TRNC | 133 | 14.74 | 2.97 | 5 | 20 | 2.980 | 0.031 * | 1–2 |
| | TR | 110 | 13.57 | 3.24 | 6 | 19 | | | |
| | TRNC + TR | 80 | 14.21 | 2.81 | 6 | 19 | | | |
| | Other | 34 | 14.26 | 2.97 | 7 | 19 | | | |
| In-organisation incentive structure of | TRNC | 133 | 14.56 | 2.72 | 5 | 20 | 0.512 | 0.674 | |
| | TR | 110 | 14.32 | 2.71 | 7 | 19 | | | |
| | TRNC + TR | 80 | 14.75 | 2.82 | 6 | 20 | | | |
| | Other | 34 | 14.85 | 3.50 | 5 | 20 | | | |
| To knit belonging and | TRNC | 133 | 20.31 | 3.62 | 9 | 29 | 0.165 | 0.920 | |
| | TR | 110 | 20.11 | 3.13 | 12 | 27 | | | |
| | TRNC + TR | 80 | 20.28 | 3.51 | 11 | 26 | | | |
| | Other | 34 | 19.91 | 3.10 | 11 | 27 | | | |
| Place of punishment | TRNC | 133 | 11.22 | 1.98 | 4 | 15 | 1.827 | 0.142 | |
| | TR | 110 | 11.20 | 1.83 | 6 | 15 | | | |
| | TRNC + TR | 80 | 10.64 | 2.12 | 6 | 15 | | | |
| | Other | 34 | 10.85 | 1.96 | 8 | 15 | | | |

* $p < 0.05$.

It was determined that there was no statistically significant difference between the scores of academicians in sub-dimensions of commitment to the organisation, organisational climate and control of individuals, organisation regulation and management authority ($p > 0.05$). According to their nationalities, there is a statistically significant difference in the scores of academicians from the sub-dimension of the organisational incentive structure. It was found that only the TRNC nationals had higher scores from the internal incentive structure sub-dimension than those of only Turkish nationals ($p < 0.05$).

When the findings related to the comparison of the scores of the Academicians in the Organisational Discipline Scale according to the education level given in Table 12. According to the educational status of the academicians, there was a statistically significant

difference between the scores of the organisational communication and organisational rules, commitment to the organisation and organisational incentive structure sub-dimensions in the Organisational Discipline Scale ($p < 0.05$). The scores of academicians with a doctorate education in these sub-dimensions were found to be significantly higher than those with a master's education.

**Table 12.** Comparison of academicians' scores from the organisational discipline scale according to their educational status ($n = 357$).

|  | Education Status | *n* | $\bar{x}$ | s | t | *p* |
|---|---|---|---|---|---|---|
| Internal communication and organisational rules | Post Graduate | 85 | 22.07 | 3.09 | −15.272 | 0.000 * |
|  | Doctorate | 272 | 30.88 | 5.02 |  |  |
| To knit | Post Graduate | 85 | 7.14 | 1.85 | −4.378 | 0.000 * |
|  | Doctorate | 272 | 8.65 | 3.00 |  |  |
| loyalty | Post Graduate | 85 | 12.32 | 3.35 | −7.009 | 0.000 * |
|  | Doctorate | 272 | 14.81 | 2.69 |  |  |
| In-organisation incentive structure | Post Graduate | 85 | 14.27 | 2.52 | −1.066 | 0.287 |
|  | Doctorate | 272 | 14.64 | 2.90 |  |  |
| Belonging to the organisation | Post Graduate | 85 | 19.84 | 3.44 | −1.140 | 0.255 |
|  | Doctorate | 272 | 20.32 | 3.38 |  |  |
| and place of punishment | Post Graduate | 85 | 11.14 | 2.02 | 0.501 | 0.617 |
|  | Doctorate | 272 | 11.02 | 1.96 |  |  |

\* $p < 0.05$.

According to the educational status of the academicians included in the study, there is no statistically significant difference between the scores of the sub-dimensions of belonging to the organisation and the place of punishment, organisational regulation and management authority, organisational climate and control of individuals ($p > 0.05$).

Table 13 contains the results of variance analysis applied to compare the academicians' scores in the study according to their professional seniority from the Organisational Discipline Scale.

**Table 13.** Comparison of academicians' scores from the organisational discipline scale according to their professional seniority ($n = 357$).

|  | Seniority | *n* | $\bar{x}$ | s | Bottom | Top | F | *p* | Variation |
|---|---|---|---|---|---|---|---|---|---|
| In-organisation Contact and organisation | 5 years and below | 101 | 22.20 | 3.17 | 15 | 32 | 360.655 | 0.000 * | 1–2 |
|  | 6–15 years | 102 | 27.62 | 4.40 | 14 | 38 |  |  | 1–3 |
|  | 16 years and above | 154 | 33.86 | 2.81 | 24 | 40 |  |  | 2–3 |
| rules | 5 years and below | 101 | 7.84 | 2.63 | 3 | 15 | 4.961 | 0.007 * | 1–2 |
|  | 6–15 years | 102 | 7.92 | 2.61 | 3 | 15 |  |  | 2–3 |
|  | 16 years and above | 154 | 8.82 | 3.04 | 3 | 15 |  |  |  |
| To knit loyalty | 5 years and below | 101 | 12.30 | 3.31 | 5 | 20 | 34.502 | 0.000 * | 1–2 |
|  | 6–15 years | 102 | 15.32 | 2.20 | 9 | 20 |  |  | 1–3 |
|  | 16 years and above | 154 | 14.74 | 2.78 | 6 | 20 |  |  |  |
| In-organisation incentive | 5 years and below | 101 | 14.28 | 2.53 | 8 | 19 | 0.683 | 0.506 |  |
|  | 6–15 years | 102 | 14.67 | 2.71 | 8 | 20 |  |  |  |
|  | 16 years and above | 154 | 14.66 | 3.05 | 5 | 20 |  |  |  |
| structure of | 5 years and below | 101 | 19.70 | 3.50 | 9 | 26 | 1.566 | 0.210 |  |
|  | 6–15 years | 102 | 20.32 | 3.29 | 13 | 29 |  |  |  |
|  | 16 years and above | 154 | 20.45 | 3.39 | 11 | 27 |  |  |  |
| To knit | 5 years and below | 101 | 11.16 | 1.92 | 4 | 15 | 0.448 | 0.640 |  |
|  | 6–15 years | 102 | 10.90 | 2.17 | 6 | 15 |  |  |  |
|  | 16 years and above | 154 | 11.07 | 1.87 | 6 | 15 |  |  |  |

\* $p < 0.05$.

According to their professional seniority, there is a statistically significant difference between the academicians' scores in the study from the intra-organisational communication and organisational rules sub-dimension ($p < 0.05$). Academicians with professional seniority of 16 years or more received higher scores from the intra-organisational communication and organisation rules sub-dimension than academicians with 5 years or less of professional seniority and academicians with 6-15 years of professional seniority. Also, the scores of the academicians with professional seniority between 6–15 years were higher than those with 5 years or less professional seniority in this sub-dimension.

In addition, the difference between the scores of the academicians participating in the study from the sub-dimension of commitment to organisation according to their professional seniority was statistically significant ($p < 0.05$). Academicians with professional seniority of 16 years or more obtained higher scores from the sub-dimension of commitment to the organisation than academicians with professional seniority of 5 years or less and academicians with professional seniority of 6–15 years.

According to their professional seniority, the scores of the academicians from the sub-dimension of the organisational incentive structure were statistically significantly different ($p < 0.05$). The scores of academicians who have 5 years or less professional seniority are lower than the other academicians. There was no statistically significant difference between the scores of the academicians included in the study from the sub-dimensions of belonging to the organisation and the place of punishment, organisational regulation and managerial authority, organisational climate, and individual supervision ($p > 0.05$).

According to the title of the academicians included in the study, the findings obtained from the variance analysis applied to compare the scores obtained from the Organisational Discipline Scale are presented in Table 14. There is a statistically significant difference between the academicians' scores in the study from the intra-organisational communication and organisational rules sub-dimension ($p < 0.05$). It was determined that the academicians with the title of specialist and doctor got lower scores in the intra-organisational communication and organisational rules sub-dimension than the academicians with the title of assistant professor, associate professor and professor. Besides, it was observed that the academicians with the title of assistant professor got lower scores in the intra-organisational communication and organisation rules sub-dimension than the academicians with the title of professor.

The difference between the academicians' scores in the study from the sub-dimension of commitment to the organisation was statistically significant ($p < 0.05$). Academicians with the title of specialist received lower scores in the sub-dimension of commitment to the organisation than the academicians with the title of doctor and professor. Besides, the scores of the academicians with the title of assistant professor are lower than the academicians with the title of doctor.

According to age groups, the scores of the academicians included in the study from the sub-dimension of the organisational incentive structure differed statistically significantly according to age groups ($p < 0.05$). It has been determined that the scores of the academicians who have the title of specialist and doctor in the sub-dimension of the organisational incentive structure are lower than the academicians who have the titles of assistant professor and associate professor. There was no statistically significant difference between the scores of the academicians from the sub-dimensions of belonging to the organisation and the place of punishment, organisational regulation and executive authority, organisational climate and people's control ($p > 0.05$).

**Table 14.** Comparison of academicians' scores on the Organisational Discipline Scale (*n* = 357).

| | Title | *n* | $\bar{x}$ | s | Bottomm | Topp | F | *p* | Variationn |
|---|---|---|---|---|---|---|---|---|---|
| In-house communication and organisation rules | Expert | 85 | 22.07 | 3.09 | 16 | 38 | 168.371 | 0.000 * | 1–3/1–4 |
| | Doctor | 32 | 22.81 | 4.38 | 14 | 31 | | | 1–5 |
| | Assistant Assoc. | 134 | 30.35 | 3.94 | 21 | 37 | | | 2–3/2–4 |
| | Associate professor | 45 | 32.53 | 3.35 | 25 | 40 | | | 2–5 |
| | Professor | 61 | 35.03 | 2.55 | 27 | 39 | | | 3–5 |
| To knit loyalty | Expert | 85 | 7.14 | 1.85 | 3 | 13 | 16.322 | 0.000 * | 1–2 |
| | Doctor | 32 | 10.53 | 3.06 | 3 | 15 | | | 1–5 |
| | Assistant Assoc. | 134 | 7.70 | 2.47 | 3 | 15 | | | 2–3 |
| | Associate professor | 45 | 8.69 | 3.14 | 3 | 15 | | | |
| | Professor | 61 | 9.70 | 3.16 | 3 | 15 | | | |
| In-house | Expert | 85 | 12.32 | 3.35 | 5 | 19 | 24.083 | 0.000 * | 1-3 |
| | Doctor | 32 | 13.03 | 3.52 | 8 | 20 | | | 1–4 |
| | Assistant Assoc. | 134 | 15.54 | 1.75 | 11 | 19 | | | 2–3 |
| | Associate professor | 45 | 15.67 | 2.25 | 10 | 20 | | | 2–4 |
| | Professor | 61 | 13.51 | 3.27 | 6 | 19 | | | |
| incentive structure of | Expert | 85 | 14.27 | 2.52 | 8 | 19 | 0.534 | 0.711 | |
| | Doctor | 32 | 14.88 | 2.60 | 10 | 20 | | | |
| | Assistant Assoc. | 134 | 14.75 | 2.80 | 8 | 20 | | | |
| | Associate professor | 45 | 14.53 | 3.21 | 5 | 19 | | | |
| | Professor | 61 | 14.38 | 3.07 | 5 | 20 | | | |
| Organisational being and place of punishment | Expert | 85 | 19.84 | 3.44 | 13 | 26 | 2.024 | 0.091 | |
| | Doctor | 32 | 19.56 | 3.71 | 9 | 25 | | | |
| | Assistant Assoc. | 134 | 20.22 | 3.29 | 12 | 29 | | | |
| | Associate professor | 45 | 21.42 | 3.04 | 13 | 26 | | | |
| | Professor | 61 | 20.10 | 3.52 | 11 | 27 | | | |
| | Expert | 85 | 11.14 | 2.02 | 4 | 15 | 0.630 | 0.642 | |
| | Doctor | 32 | 11.16 | 1.94 | 7 | 15 | | | |
| | Assistant Assoc. | 134 | 10.89 | 2.02 | 6 | 15 | | | |
| | Associate professor | 45 | 11.38 | 1.76 | 7 | 15 | | | |
| | Professor | 61 | 10.97 | 1.97 | 6 | 15 | | | |

* $p < 0.05$.

Table 15 shows the findings regarding comparing the scores of the academicians from the Organisational Discipline Scale according to the way they work.

**Table 15.** Comparison of the scores of the academicians obtained from the organisational discipline scale according to the way they work (*n* = 357).

| | Work Type | *n* | $\bar{x}$ | s | t | *p* |
|---|---|---|---|---|---|---|
| Internal communication and organisational rules | Full time | 262 | 30.01 | 5.35 | 6.897 | 0.000 * |
| | Part-time | 95 | 25.38 | 6.27 | | |
| To knit | Full time | 262 | 8.21 | 2.86 | −0.869 | 0.386 |
| | Part-time | 95 | 8.51 | 2.79 | | |
| Loyalty | Full time | 262 | 15.59 | 1.96 | 21.418 | 0.000 * |
| | Part-time | 95 | 10.42 | 2.15 | | |
| In-house incentive structure | Full time | 262 | 14.54 | 2.91 | −0.183 | 0.855 |
| | Part-time | 95 | 14.60 | 2.54 | | |
| Belonging to the organisation | Full time | 262 | 20.21 | 3.52 | 0.076 | 0.939 |
| | Part-time | 95 | 20.18 | 3.05 | | |
| and the place of punishment | Full time | 262 | 10.97 | 1.99 | −1.184 | 0.237 |
| | Part-time | 95 | 11.25 | 1.93 | | |

* $p < 0.05$.

The difference between the scores of the academicians in the organisational communication and organisational rules and the organisational incentive structure sub-dimensions in the Organisational Discipline Scale was statistically significant ($p < 0.05$). Full-time academicians had higher organisational communication and organisational rules and organisational incentive structure sub-dimensions than academicians working part-time.

According to the study style of the academicians, there was no statistically significant difference between the sub-dimensions of commitment to the organisation, belonging to the organisation and the place of punishment, organisational regulation and executive authority, organisational climate and control of individuals ($p > 0.05$).

## 4. Discussion

The COVID-19 outbreak, which affected the world, also had an impact on sustainability in education. Despite the difficulties in terms of hardware, technology literacy and connectivity, the training started online quickly. It also affected the functioning and management dimensions of higher education institutions. This process was tried to overcome the minor loss compared to face-to-face training by monitoring the number of courses that the employees of higher education institutions were inspected and the content they prepared. It has been observed that organisational discipline maintains its importance in the online period and the supervision of employees continues.

In comparing the participants according to the gender variable, as a result of the organisational discipline application, a difference was found only in the sixth sub-dimension as a result of the evaluation made in six sub-dimensions. This sub-dimension named as factor six was named as "Organisational Climate and control of individuals". This sub-dimension consists of three items: "Organisational discipline is the control and control of every action and behaviour of individuals.", "Organisational discipline creates an environment where people can easily explain their feelings and thoughts." Organisational discipline environments that enable individuals to develop their self-management and control skills create in the form. According to the answers given to these items, it was determined that the scores of female academicians in the organisational climate and control of individuals sub-dimension were higher than male academicians. This reveals that women academics are more affected by the higher education institution they work in terms of organisational climate and individuals' control [26].

Differences emerged in the first, second and third sub-dimensions as a result of the implementation of organisational discipline in higher education institutions. Factor 1, factor 2 and factor 3, Factor 1 is named as "Internal communication and organisational rules", factor 2 as "Commitment to the organisation," and factor 3 as "Internal incentive structure". Considering the answers given by the participants to the items in all three sub-dimensions, with the increase of age, individuals; It is observed that the perception of intra-organisational communication, organisational rules, commitment to the organisation and the structure of incentives within the organisation has increased. These findings are supported by some studies [27]. Concluded that the perception of organisational commitment is higher with increasing age [28]. Career plans change as age increases. In this process, commitment to the organisation, which is a tool for realizing employees' plans, can be further strengthened [29].

According to the variable of nationality, as a result of the evaluation made in six sub-dimensions, it was determined that there were differences in the first and third sub-dimensions as a result of the implementation of organisational discipline in higher education institutions. Factor 1 of these sub-dimensions named Factor 1 and Factor 3, Factor 1 was named "Organisational Communication and Organisation Rules" and factor 3 as "Internal Incentive Structure". When the participants' responses to the items in both sub-dimensions are examined, it is seen that internal communication in nationalities affects the perception of organisational rules and organisational incentive structure. Accordingly, they are less inclined to communicate only with TRNC citizens and comply with the organisation's rules in this sense. The incentive structure, which manifests itself in

reward and punishment within the organisation, shows that it is more embraced by TRNC participants than participants from foreign countries. New approaches to management are aware of the needs and expectations of the employees; It is known that the satisfaction an individual provides from his/her job affects his/her performance level. In this respect, "encouragement and rewarding" has been the subject of many theories and research for organisations [30].

When the educational status of the participants was examined according to the organisational discipline variable, it was determined that there were differences in the first, second and third sub-dimensions as a result of the evaluation made in six sub-dimensions. Factor 1, factor 2 and factor 3, Factor 1 is named as "Internal communication and organisational rules", factor 2 as "Commitment to the organisation" and factor 3 as "Internal incentive structure". Considering the answers given by the participants to the items in each of the three sub-dimensions, it was found that those with a doctorate education were more successful in terms of intra-organisational communication, commitment and incentive structure. Although some studies show that with the increase in education level, the level of knowledge of individuals increases. Therefore, their ties to the organisation will decrease. Some studies show the opposite; In the literature, this issue could not find the answer to its fullest sense [31].

As a result of the evaluations made in six sub-dimensions, differences emerged in the first, second and third sub-dimensions as a result of the implementation of organisational discipline in higher education institutions. Factor 1, factor 2 and factor 3, Factor 1 is named as "Internal communication and organisational rules", factor 2 as "Commitment to the organisation" and factor 3 as "Internal incentive structure". Considering the answers given by the participants to the items in all three sub-dimensions, as the seniority increases, the communication in the organisation, the adoption and compliance with the rules, the commitment to the organisation and the approaches that encourage the functioning in the organisation are more adopted. This reveals that with the increase in seniority in the organisation, people's desire to work in other organisations decreases as a result of the effort they spend for that organisation and years of experience [31]. As the service period increases, the earnings and seniority that the employee earns from the organisation will increase and the desire to stay where the employees tend to protect their earnings will increase, so adaptation to the organisational order will increase [32]. The more important the organisation's priorities are for the employee and the more the employee has invested in these priorities, the more his commitment to the organisation increases and it becomes difficult to leave the organisation [33]. It takes time to transfer the employees' knowledge, experience, and expertise levels gained in their organisations to new organisations. What they gain is not very applicable in the new workplace, increasing the commitment of employees to their organisations [29].

When the title status of the participants was examined according to the organisational discipline variable, it was determined that there were differences in the first, second and third sub-dimensions as a result of the evaluation made in six sub-dimensions. Factor 1, factor 2 and factor 3, Factor 1 was named as "Internal communication and organisational rules", factor 2 as "Commitment to the organisation," and factor 3 as "Internal incentive structure". Considering the answers given by the participants to the items in each of the three sub-dimensions, it is observed that academicians with the titles of education experts and doctors are more unsuccessful in the organisation than those with the titles of assistant professor, associate professor and professor than the items in the organisational communication and organisational rules sub-dimension; Also, it has been observed that academics holding the title of assistant professor are more unsuccessful than academicians with the title of professor in the sub-dimension of organisational communication and organisational rules. Accordingly, with the increasing title, people can take a more place in the higher education institution where they are located, and as the rewards within the reward-punishment structure will increase, their satisfaction increases and they see themselves as belonging to that institution.

As a result of the implementation of organisational discipline in higher education institutions, it was determined that there were differences in the first and third sub-dimensions as a result of the evaluation made in six sub-dimensions. Factor 1 of these sub-dimensions was named Factor 1 and Factor 3; Factor 1 was named "Organisational Communication and Organisation Rules" and factor 3 as "Internal Incentive Structure". Considering the answers given by the participants to the items in both sub-dimensions, the scores of the academicians working full-time in the intra-organisational communication and organisational rules and the internal incentive structure sub-dimensions were found to be higher than the academicians working part-time. This means that full-time employees are more successful than part-time employees in obeying the rules in the organisation, ensuring that these rules are fulfilled, working by adopting the reward penalty structure of the organisation and communicating with other people in this direction; With the increase of time spent in the organisation, the positive effects of integration with the organisation may be in question.

## 5. Conclusions and Suggestions

Organisational discipline is thought to be important in terms of ensuring sustainability in education. Accordingly, organisational discipline perceptions of higher education institution employees are important in terms of sustainable education. Ensuring discipline in the organisation is essential. However, in some crises, problems may arise. In such cases, it is the perception of discipline about the organisation that will guide the organisation and contribute to education sustainability.

The results obtained in the study are explained in a way that is consistent with the sub-goals and findings.

In the evaluation of organisational discipline in higher education institutions, according to the results compared according to the gender variable of the participants, it was found that the answers given to the questions about organisational discipline in higher education institutions revealed that there was a difference in the sub-dimension of "organisational climate and control of individuals" compared to men. In this regard, female participants reveal that their perception of behaviour and relationship is better in their organisations. Accordingly, women participants can express themselves correctly in the organisation and manage themselves in this direction. In the study conducted by Kobbs and Arvey (1993), a statistically significant difference was found between the gender variable and the discipline within the organisation, and it was found that male nurses displayed non-disciplinary behaviour [34]. In the research of Avcı, Küçükusta and Tütüncü (2007), it was concluded that males were more likely to move away from organisational discipline than female employees [35]. Yalçınsoy found similar results in his study in 2019 [36].

In the evaluation of organisational discipline in higher education institutions, according to the results compared according to the age variable of the participants, in evaluating the organisational discipline in higher education institutions, with the increase of age; It is observed that the perception of intra-organisational communication, organisational rules, commitment to the organisation, and organisational incentive structure has increased. Accordingly, since the older people in the organisation spend more time in higher education institutions; They know the importance of organisational rules and perceive that acting accordingly can bring positive results to the organisation. However, they are open to communication within the organisation's discipline and have reached a level where they can meet the criticisms against each other positively with the increasing age. They have adopted the reward-punishment system within the organisation and realised the arrangements that benefit the organisation. The perception that the new places where they can work decreases with the ageing of the people working in the organisation are also shown as a reason that people can exhibit loyal behaviour. Türkoğlu (2011) supports this point [37]. The age variable has an effect on the discipline in the organisation [38].

According to this study, it was revealed that the participants who were only TRNC citizens according to the nationality variable were less inclined to obey the organisation rules. This can be considered the advantage of being in their own country and the predom-

inance of non-organisational communication. The incentive structure that manifests itself in the form of rewards and punishments within the organisation, its adoption by the TRNC participants more, and the adaptation process of the participants coming to the TRNC from abroad have preceded this incentive structure.

In evaluating the organisational discipline in higher education institutions, according to the educational status of the participants, doctoral graduates are more successful in evaluating organisational discipline than graduate and undergraduate graduates. In this sense, as the professionalism of individuals increases with the increase in the level of education, the adoption of organisational commitment is positively affected by the organisational communication and reward-punishment structure, and even from these individuals. However, some studies show that these relationships with increasing education levels are inverse. It reveals that with the increasing education level, the commitment to the organisation decreases. It is argued that professionals will provide this commitment due to increased professionalization with a decrease [39].

When the seniority variable of the participants is examined in the evaluation of the organisational discipline in higher education institutions, it is revealed that the organisational discipline has developed positively with seniority. It reveals that with the increase of time spent in the organisation, it is normal for the organisation to increase its commitment to the organisation and adopt the rules and incentive structure [40]. Wolff et al. (2010), concluded that as the number of years worked increased, compliance with the rules within the organisation increased and discipline was ensured [38]. Similarly, Chullen et al. (2010) found a positive relationship between the increase in the office of healthcare workers and individuals' compliance with internal discipline [41].

The title variable of the participants in the evaluation of the organisational discipline in higher education institutions is examined. It is revealed that organisational discipline is positively affected by the increase of the title. Accordingly, with the increasing title, people can take a more place in the higher education institution where they are located, and as the rewards within the reward-punishment structure will increase, their satisfaction increases and they see themselves as belonging to that institution.

Compared to the working style variable of the participants, when the working style variable of the participants in the evaluation of the organisational discipline in higher education institutions is examined, it has been found that full-time employees are more successful in organisational discipline than those working part-time. This is due to the positive effects of integrating with the organisation as the time spent in the organisation increases.

With the emphasis on online collaboration and modelling, the institution can be encouraged at the sustainability of education. Providing training to young academicians in higher education institutions on organisation, communication and adherence to organisational rules; communication and compliance with organisational rules should be encouraged by making them feel a part of the organisation. Encouraging the institution's employees to access and use data sources and ensuring their readiness will positively affect sustainable education.

All stakeholders should be informed about the results obtained with learning analytics for the development of sustainable education. With the strategies to be prepared accordingly, learners' scope of learning outcomes can be determined and a road map can be drawn. Online education is expected to continue in the future. Knowing the organisational discipline perceptions of the institution's employees and integrating into the system accordingly and taking advantage of its advantages will be important in the future in terms of the sustainability of education. According to the results obtained as a result of the research, some suggestions can be made.

This study stated that women working in higher education institutions for sustainability in education were more affected by the organisational climate and the sub-dimension of people's control. In this sense, female employees should be reminded frequently that they are professional in their organisation and should be more encouraged in their speeches or

recommendations on business-related issues. Besides, the institution can be encouraged at the point of sustainability of education regarding online cooperation and modelling.

In the research, education should be given to young academicians in higher education institutions about commitment to organisation, communication and organisation rules.

In terms of intra-organisational communication and compliance with organisational rules, TRNC citizens should be encouraged to perceive the organisation as a whole and share that they are a part of it and have the responsibility in this direction.

The internal incentive structure should be of a size that all people can envy and meet their needs. Graduate academicians should be encouraged to pursue a doctorate, as the academicians working in higher education institutions adapt better to the sub-dimensions of organisational discipline.

The positive situations occurring in the organisational discipline sub-dimensions of the organisational commitment, communication and incentive structure with the increase in the seniority of employees in higher education institutions; The rewards to be given to them should be increased for them to be adopted by lower seniors and the awareness that they are valuable for that organisation can be developed by asking them to contribute more to the organisation. Encouraging organisation employees to access and use data sources and ensure their readiness will positively affect sustainable education.

Since the title affects the sub-dimensions of the organisational discipline, other academicians can be supported in keeping up with the organisational discipline with different specialities that can be created within the higher education institution or other responsibilities that can be assigned to them.

Part-time employees should also recognise that they are a part of this organisation and raise awareness of the need to comply with these rules, or the organisation should be strengthened by other people who can replace them. Rules regarding organisational discipline should be understandable and clearly stated.

Impunity discipline should be developed. Individuals working under the organisational discipline should be motivated by awards.

For sustainability in education, there should be evidence-based evaluations by making learner-oriented process evaluations. All stakeholders should be informed about the results obtained with learning analytics to develop sustainable education. With the strategies to be prepared accordingly, learners' learning outcomes can be determined and a road map can be drawn. Online training is expected to continue in the future. Therefore, integrating into the system and taking advantage of its advantages will be important in the future in terms of education sustainability.

In the name of sustainability of education, managers should lead, have a vision about online work and guide employees. While determining the leader's vision, she should know the employees' perceptions in the dimension of organisational discipline and evaluate accordingly.

Figure 2 shows that the Organisational Discipline Scale is a valid measurement tool with its 30-item six-factor structure, so it can be recommended to be used for similar studies.

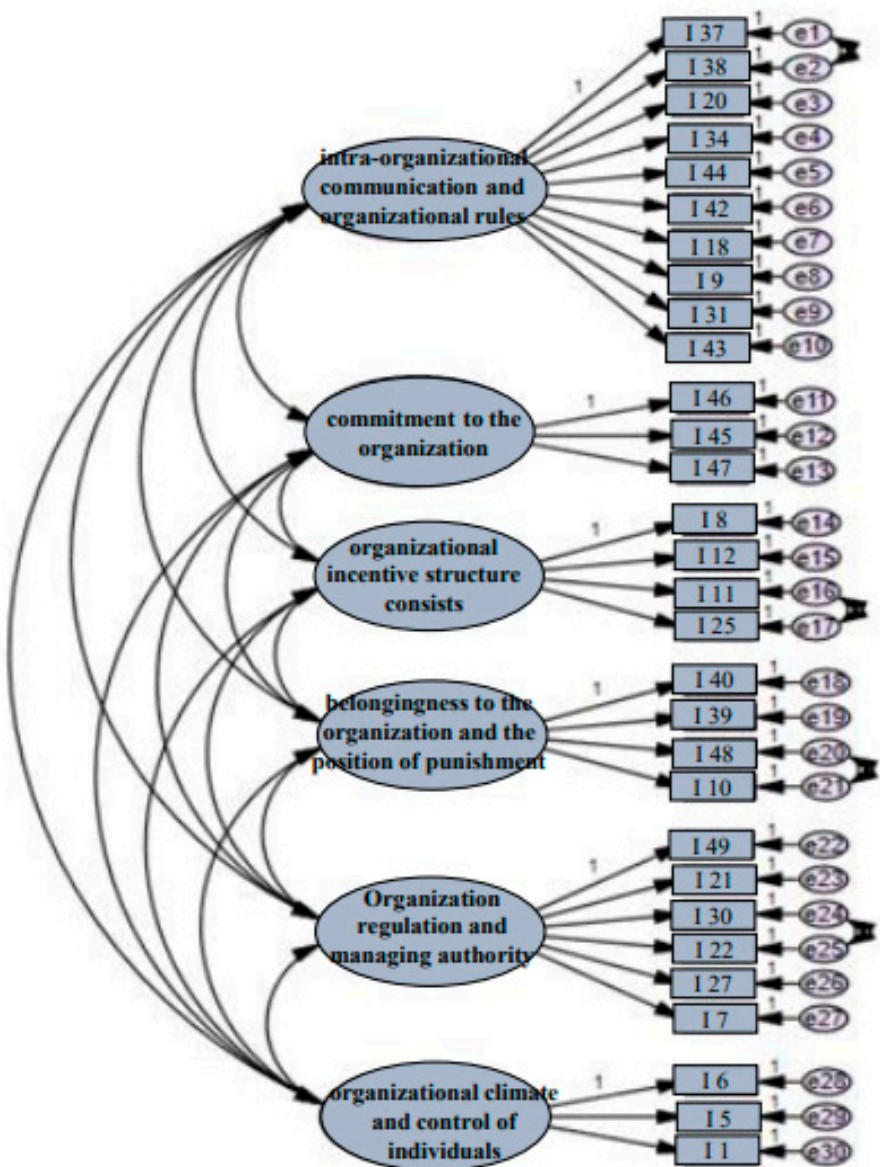

**Figure 2.** Confirmatory factor analysis path diagram of the organisational discipline scale.

**Author Contributions:** Conceptualization, R.Y., G.D., Z.A., M.A. and E.O.; methodology, R.Y.; software, R.Y.; validation, G.D.; formal analysis, R.Y.; investigation, G.D., F.A. and Z.A.; resources, G.D., Z.A., M.A. and E.O.; data curation, R.Y.; writing—original draft preparation, R.Y. and F.A.; writing—review and editing, F.A. and E.O.; visualization, R.Y.; supervision, G.D.; project administration, F.A.; funding acquisition, R.Y. All authors have read and agreed to the published version of the manuscript.

**Funding:** This research received no external funding.

**Institutional Review Board Statement:** The study was conducted according to the following guidelines: Statement Approved by the Near East University Scientific Research Ethics Committee (IR 455/2020; approval date: 24 April 2020).

**Informed Consent Statement:** Written informed consent has been obtained from the patient(s).

**Data Availability Statement:** MDPI Research Data Policies rules have been checked. The study is prepared following these rules.

**Conflicts of Interest:** The authors declare no conflict of interest.

**Appendix A**

\* Organisational discipline sub-dimensions and items in sub-dimensions. Participants answered these items, which were applied to them as a questionnaire, by ticking the appropriate option from the options of strongly disagree, disagree, neither agree nor disagree, agree, strongly agree.

- **Internal communication and organisational rules:**

Item 37: In organisational discipline, people can criticise each other.

Item 38: Within the organisational discipline, each individual has similar working conditions and opportunities.

Item 20: The method of reward and punishment in organisational discipline causes the self-control ability of individuals to weaken.

Item 34: In terms of organisational discipline, people are constantly compared to each other.

Item 44: Due to your satisfaction with the organisational discipline of your university, you think you owe a lot to the organisation.

Item 42: You are satisfied with your university's understanding of organisational discipline.

Item 18: It is easy for people who follow the rules to rise in organisational discipline.

Item 9: To maintain organisational discipline, all members of the organisation must be punished for an unknown issue.

Item 31: Penalties given to individuals due to organisational discipline are not kept secret.

Item 43: Within the framework of your university's organisational discipline, you want to perceive and solve the problems faced by the organisation as if they were your own.

- **Commitment to the organisation:**

Item 46: Since you are satisfied with the organisational discipline of your university, you will not accept another University that offers you a better position.

Item 45: Due to the tough organisational discipline of your university, you are thinking of leaving.

Item 47: Even if you are not satisfied with the organisational discipline of your university, you cannot leave your university because you have no other choice.

- **Intra-organisational incentive structure:**

Item 8: In organisational discipline, the reward is an important factor for people to engage in positive behaviour.

Item 12: Rewards given in organisational discipline reinforce positive behaviours.

Item 11: In organisational discipline, individuals are allowed to defend themselves before being punished.

Item 25: To ensure organisational discipline, exemplary people who follow the rules are rewarded for all to see.

- **Belonging to the organisation and the place of punishment:**

Item 40: Within organisational discipline, people develop emotional commitment.
Item 39: In organisational discipline, people feel like one of the family.
Item 48: Punishment is often used to maintain organisational discipline.
Item 10: Punishment is the last resort in organisational discipline.

- **Organisation regulation and managerial authority:**

Item 49: To ensure organisational discipline, disciplinary rules are frequently reminded to prevent undesirable behaviours.

Item 21: To ensure organisational discipline, the rules of the regulation should be reminded to people every year.

Item 30: Evaluation of people and their work is done consistently and fairly to ensure organisational discipline.

Item 22: The mistakes that people who are successful in organisational discipline make from time to time are ignored.

Item 27: To ensure organisational discipline, the tolerant and mild nature of the people who will determine the punishment negatively affects the provision of discipline.

Item 7: Organisational discipline is to ensure that people enter and conform to the desired patterns.

- **Organisational climate and control of individuals:**

Item 6: Organisational discipline creates environments that enable individuals to develop self-management and control skills.

Item 5: Organisational discipline creates an environment where people can freely express their feelings and thoughts.

Item 1: Organisational discipline is the supervision and control of every movement and behaviour of individuals.

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
