# Peer review of "Sustainability in Education: A Scale on Perceptions of Organisational Discipline Related to the COVID-19 Period"

_sustainability, doi:10.3390/su13158343_

Round 1

Reviewer 1 Report

It is an interesting paper and accomplished the scientific requirements to be published in the journal. Nonetheless, there are some minor recommendations to improve it as describing general and specific aims that complete the researching questions; clarifying the design of the study (type of method, methodology...); designing of the scale, how has it been designed?. Is it "ad hoc" or adapted from others? (explaining the structure and their field of knowledge); joining the section 5 suggestions by "conclusions" and focusing on previous aims. 

Author Response

Response to Reviewer 1 Comments

Point 1: Clarifying the design of the study (type of method, methodology...); designing of the scale, how has it been designed?. Is it "ad hoc" or adapted from others? (explaining the structure and their field of knowledge)

Response 1: The study was based on a survey for the quantitative data, which is one of the tech-niques generally used in the scanning model, was used. The scale was prepared in a 5-point Likert type. Accordingly, the participants answered the questionnaire as strongly disagree, disagree, neither agree nor disagree, agree, strongly agree. The questionnaires were delivered to a group of 357 randomly selected personnel working in higher education institutions during the pandemic period.
 In the study, the organizational discipline scale was used with personal information form as a quantitative data collection tool in higher education institutions. A question bank was created by making a literature review on the subject and determining which questions or items would be appropriate for the scale subject. When designing questions to be understandable care was taken. Expert opinion was requested for the questions in the created question bank. Experts on the necessity, clarity and specificity of questions made its assessment. The scale, which was shaped according to expert opinion, applied to the sample group. For the validity and reliability study of the draft Organizational Discipline Scale developed by the researcher based on the literature review, the opinions of Turkish teachers and experts of the subject and the construct validity of the scale were examined first. While examining the construct validity of the Organizational Discipline Scale, Explanatory Factor Analysis (EFA) and Confirmatory Factor Analysis (CFA) was used to determine the scale's factor structure.

Point 2: Joining the section 5 suggestions by "conclusions" and focusing on previous aims. 

Response 2: Section 6 has been removed. The conclusions in section 6 are in the same section as the recommendations in section 5.

Reviewer 2 Report

The article shows a study carried out through surveys. A statistical study is carried out in which it compares different dimensions in the structure of organizations, as a result of COVID-19. Study sub-dimensions of the scale: "Internal communication and organizational rules," "Commitment to the Organization," "Organizational incentive structure," "Organization and the place of punishment," "Organization regulation and management authority," "Organizational climate and control of individuals."

Research questions (lines from 137 to 146) are not of significant interest.

It would be interesting to include a copy of the survey carried out, with each of the questions that were included in it.

The figures are of poor quality. It is recommended to improve the quality of the images included in the job.

On line 382 there is an error. Where it says “40-40”, it should say “40-49”.

Suggestions are very general and no concrete suggestions are given to change the detected problems or are not suggestions that can be carried out.

Some conclusions are obvious, such as those indicated in lines from 698 to 704. They are conclusions that contribute little to the content of the article. 

The work has an extensive bibliography, quite current.

Author Response

Response to Reviewer 2 Comments

Point 1: It would be interesting to include a copy of the survey carried out, with each of the questions that were included in it.

Response 1: The items in the questionnaire are listed under the sub-dimensions and added to the end of the study as Annex 1.

Point 2: On line 382 there is an error. Where it says “40-40”, it should say “40-49”.

Response 2: The said place has been changed to 40-49.

The said place has been changed to 40-49.

Reviewer 3 Report

The topic is very interesting, however I suggest the following additions:
- strengthen the theoretical basis of the proposed scale (e.g. how were the sub dimensions of the scale defined?);
- describe in more detail how the survey was carried out: when it was carried out, how it was carried out, how the subjects analysed were selected;
- specify the content of the survey and how the questions posed to the 357 subjects are related to the sub-dimensions of the scale;
- strengthen the link, mentioned in the conclusions, between discipline and sustainability of the education system.

Author Response

Response to Reviewer 3 Comments

Point 1: Strengthen the theoretical basis of the proposed scale (e.g. how were the sub dimensions of the scale defined?;

Response 1: In the literature review of this study, no scale related to the perception of organizational discipline was found. Sub-dimensions were determined by examining the information related to the subject in the literature. For this purpose, a scale consisting of six sub-dimensions that will cover the perception of organizational discipline was prepared. The sub-dimensions of the scale are "Internal communication and organizational rules," "Commitment to the Organization," "Organizational incentive structure," "Organization and the place of punishment," "Organization regulation and management authority," "Organizational climate and control of individuals." For the questions representing the sub-dimensions, a question bank was created in line with the information obtained from the literature review and the scale was designed by taking expert opinions. It is aimed to fill the gap in the field with a valid and reliable organizational discipline perception scale prepared according to these sub-dimensions. 

Point 2: Describe in more detail how the survey was carried out: when it was carried out, how it was carried out, how the subjects analysed were selected;

Response 2: The study was based on a survey for the quantitative data, which is one of the tech-niques generally used in the scanning model, was used. The scale was prepared in a 5-point Likert type. Accordingly, the participants answered the questionnaire as strongly disagree, disagree, neither agree nor disagree, agree, strongly agree.   The questionnaires were delivered to a group of 357 randomly selected personnel working in higher education institutions during the pandemic period. 

Point 3: specify the content of the survey and how the questions posed to the 357 subjects are related to the sub-dimensions of the scale;

Response 3: In the literature review of this study, no scale related to the perception of organizational discipline was found. Sub-dimensions were determined by examining the information related to the subject in the literature. For this purpose, a scale consisting of six sub-dimensions that will cover the perception of organizational discipline was prepared. The sub-dimensions of the scale are "Internal communication and organizational rules," "Commitment to the Organization," "Organizational incentive structure," "Organization and the place of punishment," "Organization regulation and management authority," "Organizational climate and control of individuals." For the questions representing the sub-dimensions, a question bank was created in line with the information obtained from the literature review and the scale was designed by taking expert opinions. It is aimed to fill the gap in the field with a valid and reliable organizational discipline perception scale prepared according to these sub-dimensions. To achieve this aim, the following research questions were created and answers were sought. As a sample group, 357 people to whom the questionnaire was applied are higher education employees. These people, who were randomly selected and surveyed, were determined considering that they work in an area where organizational discipline is important in terms of sustainable education.

Point 4: Strengthen the link, mentioned in the conclusions, between discipline and sustainability of the education system.

Response 4: Organizational discipline is thought to be important in terms of ensuring sustainability in education. Accordingly, organizational discipline perceptions of higher education institution employees are important in terms of sustainable education. Ensuring discipline in the organization is essential. However, in some crisis situations, problems may arise. In such cases, it is the perception of discipline about the organization that will guide the organization and contribute to the sustainability of education.

Round 2

Reviewer 2 Report

The statistical treatment is interesting and includes a lot of data.
The topic could have been approached differently to make it more interesting.